# Exogenous Application of Indol-3-Acetic Acid and Salicylic Acid Improves Tolerance to Salt Stress in Olive Plantlets (*Olea europaea* L. Cultivar Picual) in Growth Chamber Environments

**María del Pilar Cordovilla** [1,2,*], **Carolina Aparicio** [1], **Manuel Melendo** [1] and **Milagros Bueno** [1]

1   Plant Physiology Laboratory, Department Animal Biology, Plant Biology and Ecology, Faculty of Experimental Science, University of Jaén, Paraje Las Lagunillas, E-23071 Jaén, Spain
2   Center for Advances Studies in Olive Grove and Olive Oils, Faculty of Experimental Science, University of Jaén, Paraje Las Lagunillas, E-23071 Jaén, Spain
*   Correspondence: mpilar@ujaen.es; Tel.: +34-953-212-786

**Abstract:** Salinity is one the most recurrent abiotic stresses worldwide and severely affects crop productivity in arid and semiarid environments. This research analyzed several plant growth regulators that could mitigate the effects of salinity on olive plants (*Olea europaea* L. cultivar Picual). Mist-rooted cuttings were grown in a growth chamber and pretreated with gibberellic acid ($GA_3$), indole-3-acetic acid (IAA), salicylic acid (SA), and Kinetin by foliar spraying twice a week for three weeks. At the end of the pretreatment, the plants were exposed to 100 mM and 200 mM sodium chloride (NaCl) for six weeks. The results showed that plants pretreated with the plant growth regulators significantly increased their biomass under saline conditions. In addition, IAA and SA restricted the transport of sodium ($Na^+$) ions from roots to leaves and improved the leaf potassium ($K^+$)/$Na^+$ ratio. IAA and SA favored proline, fructose, and mannitol accumulation in leaves at 100 mM and 200 mM NaCl, as did glucose at 200 mM NaCl. Salicylic acid and IAA increased pigments (chlorophylls and carotenoids) and polyamines accumulation under saline conditions. The findings of this study suggest that pretreatments with IAA and SA may be a highly effective way of increasing salt tolerance in olive plantlets.

**Keywords:** biomass; $Na^+$; $K^+$; plant growth regulators; olive; osmolytes; pigments; polyamines; pretreatment; salinity

## 1. Introduction

Soil salinity is caused by a high concentration of soluble salts that retain water in the soil and accentuate the problem of desertification. Salt affects the availability of carbon and nitrogen in the soil, and also has a bearing on various microbial processes and activities, and impacts agricultural productivity and environmental sustainability [1]. Soil salinity influences plant growth and development as a consequence of the osmotic effect, ionic toxicity, and the nutritional imbalance caused by salt in plant tissue [2,3]. In addition to natural and anthropogenic causes, climate change is leading to greater desertification and soil salinization, and increasingly less land is now devoted to agricultural production [4,5]. Currently, salinized land occupies a total of 932.2 Mha worldwide, which in Europe is mainly concentrated in the Mediterranean area [6,7]. Salinization decreases the production of food and fodder. Thus, in the twenty-first century, the development of technologies for increasing agricultural production has become a research priority aimed at providing food for the approximately 9 billion people expected to populate our planet by 2050 [5,8]. Innovative technologies, such as gene-editing, genome database information, and trans-genesis [9–11], in addition to seed priming and foliar spraying (techniques that enhance germination and growth development by activating various physiological and biochemical

processes) with compounds, such as plant growth regulators (PGRs) [12–16], could all help improve abiotic stress tolerance in both herbaceous and perennial crops.

In recent years, research aimed at improving saline tolerance in different cultivars via the foliar application of PGRs has increased [16–18]. One of the most widely used among such compounds is salicylic acid (SA) [19], and both field and laboratory works have shown that this PGR mitigates the detrimental effects of salinity by stimulating the physiological and metabolic mechanisms that improve chlorophyll content, stomatal conductance, and leaf relative water content (LRWC), thereby protecting membrane integrity in both leaves and roots [20,21]. This growth regulator also protects against oxidative damage and increases antioxidant activity and accumulations of osmolytes that could help protect photosynthetic mechanism under saline conditions [22,23]. The foliar application of gibberellins (GA$_3$) has also been explored in numerous trials as a way of increasing productivity under saline conditions. The application of GA$_3$ in *Vigna mungo* increased most measured ecophysiological and biochemical parameters and restored them to the normal values they would have had without salt [24]. Remarkable results for GA$_3$ have also been found for tomato and cucumber cultivation and have improved seedling quality [25]. In maize under salt stress, GA$_3$ application reduces oxidative stress by increasing antioxidant enzyme activity, antioxidant gene expression, and K$^+$ concentration [26]. Gibberellic acid has also been applied to woody plants where it can play an important role in reducing the negative effects of salt, probably by improving carbon assimilation and increasing the leaf chlorophyll index [27].

Less information, however, is available on the efficacy of foliar application of auxins and cytokinins to improve salt stress tolerance in agricultural crops. These phytohormones not only stimulate plant growth in general, but also improve abiotic stress tolerance [28]. Recently, it has been shown that in faba beans (*Vicia faba*) under salt stress IAA enhances the accumulation of osmolytes (soluble sugars and proteins), regulates ionic homeostasis, and improves antioxidant activity, as well as increases the number of nodules for better biological nitrogen fixation [29]. Elsewhere, kinetin (a synthetic cytokinin) has been used in combination with other phytohormones. Together, IAA and Kinetin have been found to enhance essential inorganic nutrients and maintain membrane permeability in maize (*Zea mays*) plants under field conditions [30]. Kinetin also enhanced the photosynthetic and antioxidant responses in *Nigella sativa* that help counteract the effects of salt stress [31]. Finally, in recent years, a study of sweet sorghum has shown that the application of GA$_3$, SA, and Kinetin can lead to a significant increase in photosynthetic and transpiration rates, as well as stomatal conductance [32].

In terms of endogenous plant regulators, polyamines (PAs) (putrescine (Put), spermidine (Spd), and spermine (Spm)) and ethylene are two PGRs, whose levels can increase under abiotic stress, and which can control ion homeostasis and regulate the antioxidant system. In addition, PAs can interact with other metabolic pathways by establishing hormonal crosstalk [33,34].

Little information is yet available on whether or not the foliar application of PGRs increases salt stress tolerance in woody plants. We investigated the olive tree (*Olea europaea* L. family *Oleraceae*), one of the most widespread crops in the Mediterranean areas [35], and, specifically, the Picual cultivar, one of many existing olive cultivars and widespread in the Iberian Peninsula [36]. Its fruits are appreciated for the quality of their oil, which is widely consumed in the Mediterranean diet due to the unsaturated fatty acids it contains and its antioxidant properties that have beneficial effects on health [37–39]. The Mediterranean region is characterized by its rather dry climate, especially during the summer season when drought and the use of low-quality irrigation water increase the salinity of agricultural soils [40,41]. The use of ecological strategies to achieve more salinity-tolerant cultivars in light of current climate change will improve crop yields and quality.

Tolerance to salinity can be achieved by stimulating (i) plant water content to increase cell elongation and plant growth, (ii) ion exclusion mechanisms in roots, (iii) decreased Na$^+$ and Cl$^-$ transport to aerial parts, (iv) the accumulation of compatible solutes (osmolytes) for

better osmotic adjustment and of pigments for better photosynthetic efficiency, and (v) an increase in endogenous stress-related PGRs (PAs and ethylene) [27,42–44]. The aim of this study was to determine which pre-treatments with PGRs (GA$_3$, IAA, SA and Kinetin) are most effective in improving the above-mentioned parameters and increasing the resistance of the *Olea europaea* cultivar Picual to NaCl.

## 2. Materials and Methods

### 2.1. Plant Material and Growth Conditions

One hundred and twenty uniform mist-rooted olive Picual cuttings (12-cm shoot length; three-months-old) (Viveros Jarico SL, Almería, Spain; GPS location: 37°17′6.36″ N, 1°52′8.759″ W) were transplanted to 1-L plastic pots containing a sand-perlite mixture (1:3, v/v) and placed in a growth chamber under the following conditions: a 16–8 h light-dark cycle, 25–20 °C day-night temperature, relative humidity 55–75%, and photosynthetic photon flux density (PPFD) (400–700 nm) of 500 µmol m$^{-2}$ s$^{-1}$ (Sylvania Cool White and Osram Dulux Superstar lamps; Osram Sylvania Inc., Danvers, MA, USA). Plants were irrigated three times per week with 100 mL of half-strength Hoagland's solution [45]. After four weeks acclimation, plants were randomly separated into five groups (24 plants per group) to be pretreated with the plant growth regulators (PGRs):

- Group 1: each plant was sprayed with 25 mL of distilled water (no PGR pretreatment).
- Group 2: each plant was sprayed with 25 mL of gibberellic acid (GA$_3$) (1 µM).
- Group 3: each plant was sprayed with 25 mL of indol-3-acetic acid (IAA) (1 µM).
- Group 4: each plant was sprayed with 25 mL of salicylic acid (SA) (0.5 mM).
- Group 5: each plant was sprayed with 25 mL of Kinetin (1 µM).

The PGR concentrations used were based on previous experiments and information given in the literature [19,41,46–49]. Kinetin was dissolved in 0.5 N hydrochloric acid, IAA in 1N sodium hydroxide, and salicylic acid in 97% sulphuric acid, and then diluted to the final concentration by adding distilled water. GA$_3$ was dissolved in distilled water.

These pretreatments were performed twice a week over a period of three weeks. Six randomly chosen plants per group were harvested to assess the initial biomass and ion quantification, the data from which were used to calculate the net translocation. Subsequently, the remaining plants were divided into three groups of six plants, to which three concentrations of NaCl were applied (0, 100, and 200 mM) [27]. NaCl was added to the nutrient solution incrementally over an eight-day period to avoid osmotic shock to the plants (25 mM NaCl day$^{-1}$). Plants were harvested six weeks after the start of the saline treatment. Six replicates of one plant were harvested per pretreatment with PGRs and NaCl concentration. A schematic representation of the experiment is given in Figure 1.

The dry weight of roots, stems, and leaves, Na$^+$ and K$^+$ levels, Na$^+$/K$^+$ ratio, and the net translocation rates of these ions to the leaves of olive plantlets were analyzed. We also evaluated the leaf relative water content, photosynthetic pigments, proline, alcohol-soluble sugars, sugar alcohols, starch, free polyamines, and ethylene in leaves given that the highest metabolite concentrations were found in this organ.

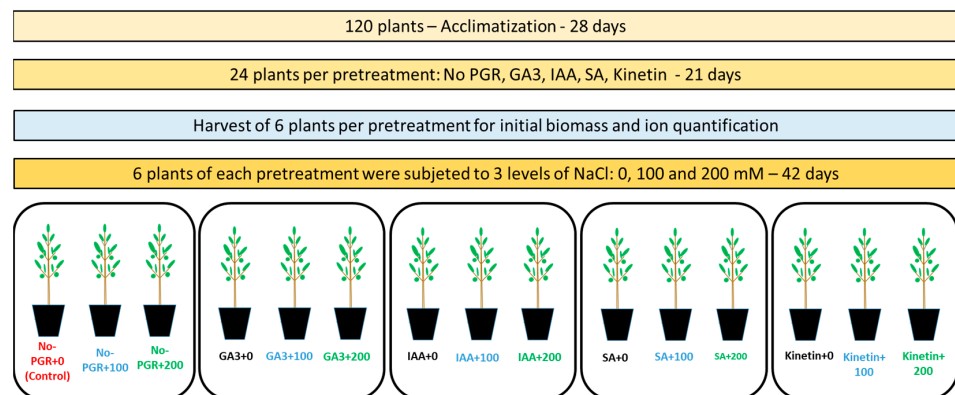

**Figure 1.** Schematization of the experiment. Non-pretreated plants with no NaCl (No-PGR; control); non-pretreated plants with 100 mM NaCl (No-PGR+100); non-pretreated plants with 200 mM NaCl (No-PGR+200); pretreated with gibberelic acid and no NaCl (GA$_3$+0); pretreated with gibberelic acid and with 100 mM NaCl (GA$_3$+100); pretreated with gibberelic acid and with 200 mM NaCl (GA$_3$+200); pretreated with indol-3-acetic acid and no NaCl (IAA+0); pretreated with indol-3-acetic acid and with 100 mM NaCl (IAA+100); pretreated with indol-3-acetic acid and with 200 mM NaCl (IAA+200); pretreated with salicylic acid and no NaCl (SA+0); pretreated with salicylic acid and with 100 mM NaCl (SA+100); pretreated with salicylic acid and with 200 mM NaCl (SA+200); pretreated with Kinetin and no NaCl (Kinetin+0); pretreated with Kinetin and with 100 mM NaCl (Kinetin+100); pretreated with Kinetin and with 200 mM NaCl (Kinetin+200).

### 2.2. Determination of Growth Parameters and Leaf Relative Water Content

When harvested, plants were gently removed from the substrate, washed with distilled water and dried between two layers of filter paper. The fresh material was used to measure ethylene production in all treatments. Roots, stems, and leaves from both control and PGR-treated plants (with salt and without salt) were dried at 70 °C for 72 h in a forced-air oven and the dry weight (DW) of the different organs was determined. The leaf relative water content (LRWC) was calculated using the following formula:

$$LRWC \ (\%) = (FW - DW)/(TW - FW) \times 100$$

where FW is the fresh weight, DW the dry weight, and TW the fresh weight at full turgor, measured after the immersion of the leaf petioles in demineralized water for 48 h in the dark at 4 °C [50].

### 2.3. Ion Quantification and Net Translocation

The Na$^+$ and K$^+$ concentrations in leaves and roots were measured with an emission-absorption spectrophotometer (Perkin Elmer AAnalyst 800, Shelton, CT, USA) after the tissue had been dry-ashed for 24 h at 450 °C and suspended in 37% hydrochloric acid (HCl). The net translocation rate from roots to leaves ($J_s$) was calculated using the following equations:

$$J_s = (MS2 - MS1)/(T2 - T1) \times (\ln(W2 - W1))/(W2/W1)$$

where $J_s$ is expressed as mmol kg$^{-1}$ root DW day$^{-1}$, MS1 and MS2 are the nutrient concentration in shoots, T2-T1 the experimental period (42 days), and W2 − W1 the difference between the root dry weight (Kg) at the end of the experimental period (W2) and prior to the salt treatments (W1) [51].

### 2.4. Determination of Photosynthetic Pigments

Leaf tissue from the middle of the main leaf axis was ground in 80% acetone [52] for chlorophyll and carotenoid determination [53]. Results were expressed as mg$^{-1}$ DW.

### 2.5. Determination of Proline

Leaf tissue was homogenized with 3% sulfosalicylic acid and centrifuged at $1000\times g$ for 5 min, and the supernatant was used for the quantification of the proline [54]. The endogenous Pro concentration was determined at 520 nm in a VARIAN spectrophotometer (Cary 4000 UV-VIS, Santa Clara, CA, USA). A standard curve with proline was used for the final calculations with quantities of between 0.5 and 20 µg. Results were expressed as µg g$^{-1}$ DW.

### 2.6. Determination of Alcohol-Soluble Sugars and Sugar Alcohols

To measure the alcohol-soluble sugars (sucrose, glucose, and fructose) and sugar alcohols (mannitol and inositol), the method used by Bartolozzi et al. [55] was followed with some modifications. Plant material (300 mg) was homogenized with 50% ethanol (1:10, w/v) containing β-phenyl-glucopyranoside (1:40, w/v) as an internal standard (5 mg g$^{-1}$ tissue). The homogenate was centrifuged at $1000\times g$ and 4 °C for 5 min. The supernatant was removed and diluted to a volume of 22.5 mL with 50% ethanol. An aliquot (1.5 mL) of the sample was dried in an air stream and treated with 0.4 mL of pyridine, 0.2 mL of hexamethyldisilazane, and 0.1 mL of trimethylchlorosilane (TMCS), and then heated at 60 °C. After 2 h, 0.3 mL of sample was injected into a HP 5890 (series II) gas chromatograph fitted with a flame ionization detector and 30 m x 0.25 mm capillary fused-silica column HP-5MS (Crosslinked 5% PME Siloxane). Injector and detector temperatures were 280 and 320 °C, respectively. Flow rates of He, H$_2$, and air were 2, 30, and 250 mL min$^{-1}$, respectively. To calculate the alcohol-soluble sugar and sugar alcohol concentrations, sucrose, glucose, fructose, mannitol, and inositol were dissolved in 1.5 mL of 50% ethanol with an imidazole buffer 0.1 M pH 7, dried in an air stream, and treated using the same procedure, with quantities of between 10 and 500 µg. Results were expressed as mg g$^{-1}$ DW.

### 2.7. Determination of Starch

To determine the starch content, plant material was homogenized three times with 95% ethanol (1:10, w/v) and centrifuged at $1000\times g$ and 4 °C for 10 min. [56]. Then, the solid fraction obtained was used for starch analysis [57]. Samples (20 mg) were heated with 2 mL of 0.1 N sodium hydroxide (NaOH) in a 50 °C water bath for 30 min with intermittent mixing. After neutralizing with 2.5 mL of 0.1 N acetic acid, 0.5 mL of a digestive enzyme mixture containing 1000 U of α-amylase (from *Bacillus licheniformis*, Sigma A-4551) and 5 U of amyloglucosidase (from *Aspergillus niger*, Sigma A-7420) in 0.05 M sodium acetate buffer (pH 5.1) was added. The combined solution was incubated for 24 h in a water bath at 50 °C. To calculate the amount of glucose hydrolysate, the digest was centrifuged at 2500 rpm for 10 min. Two mL of peroxidase-glucose oxidase/o-dianisidine reagent (PGO enzymes; Sigma P-7119) in 100 mL of distilled water mixed with 1.6 mL of o-dianisidine solution (50 mg of o-dianisidine dihydrochloride (Sigma D-3252) in 20 mL of distilled water) was added to 0.2 mL of the supernatant and left in darkness at room temperature for 45 min. The absorbance was read at 525 nm after adding 0.4 mL of 75% sulfuric acid (H$_2$SO$_4$). The amount of glucose was calculated against a glucose standard prepared in the sodium acetate buffer solution, with quantities of between 10 and 150 µg. Results were expressed as mg g$^{-1}$ DW.

### 2.8. Determination of Free Polyamines

The method used by Bueno et al. [58] was followed with minor modifications for the determination of the free polyamines (Put, Spd, and Spm). Fresh material (100 mg) was homogenized with 0.2 M perchloric acid (1:4, w/v) containing 1.6-diamino-hexane (1:1, w/v) as an internal standard (100 µg g$^{-1}$ tissue). The homogenate was centrifuged at $27,000\times g$ and 4 °C for 10 min and 0.1 mL aliquots of the supernatant were saturated with sodium carbonate and dansylated with dansyl chloride (10 mg mL$^{-1}$ in acetone). The mixture was incubated at 60 °C for 1 h before the solution of L-proline (100 mg mL$^{-1}$)

was added. After 30 min, the dansylated polyamines were extracted with toluene (HPLC grade). The toluene extract was dried under nitrogen and the residue was dissolved in acetonitrile (HPLC grade) and filtered through Millipore (Darmstadt, Germany) HV-4 filters for immediate analysis. An aliquot (0.02 mL) of the sample was injected into a reversed-phase Spheri-5 C18 ODS (8 μm, 4.6 × 220 mm) column. A Shimadzu (Kyoto, Japan) LC-10A HPLC equipped with a fluorescence spectrophotometer (the excitation and emission wavelengths were 252 nm and 500 nm, respectively) was used to quantify the dansyl derivatives [47,48,58–60]. The same procedure was applied with standards. Results were expressed as μmol g$^{-1}$ DW.

### 2.9. Determination of Ethylene Production

For the ethylene measurements, the method of Bueno et al. [59] was followed with minor modifications. Fresh leaves were transferred to 5-mL flasks containing 50 μL distilled water. The flasks were sealed with silicone-rubber stoppers and incubated for 1 h in darkness at 30 °C. Later, 1 mL gas samples were removed from the flasks and injected into a HP 5890 (series II) Hewlett Packard (Palo Alta, CA, USA) gas chromatograph fitted with a flame ionization detector and a 2 m × 4 mm stainless-steel column packed with 50–80 mesh Poropack-R. The oven temperature was 100 °C, and the N$_2$, H$_2$, and the synthetic airflow rates were 50, 86, and 400 mL min$^{-1}$, respectively. Ethylene identification was based on the retention time compared to a C$_2$H$_4$ standard (purity, 99.9%). Results were expressed as nmol g$^{-1}$ FW h$^{-1}$.

### 2.10. Statistical Analysis

To check for normality, all data were subjected to a Shapiro–Wilk test. Data sets with *p* values below the threshold of 0.05 were transformed (lg) before statistical analysis. Data were subjected to a two-way analysis of variance (effects of pretreatments with growth regulators and treatments with NaCl as fixed factors with the interaction factor). Significant differences were evaluated post-hoc using the LSD test ($p < 0.05$). All parameters studied with and without PGRs in the absence or presence of salt were compared using Pearson's correlation coefficients. All calculations, including statistical analysis, were computed using Statgraphics Centurion v. 19 (University of Jaén).

## 3. Results

For all parameters studied, the statistical analysis showed the significant effect of the interaction between the growth regulator pretreatment and NaCl treatment ($p < 0.05$).

### 3.1. Effects of the Application of Plant Growth Regulators (PGRs) on Biomass

Data for whole plants dry weight show that under non-saline conditions, the PGRs did not stimulate plant growth any more than the control plants (0 mM NaCl and no-PGR) (Figure 2). In the salt-stressed plants, dry weights (DW) were significantly lower than in the plants grown under normal conditions (control). The fall in growth increased in a dose-dependent fashion and maximum growth inhibition was observed for the most severe salt-stressed olive plants (200 mM NaCl). The greatest reductions in DW were observed in plants not pretreated with PGR: at 200 mM NaCl the whole plant DW fell by 46.47% (Figure 2), root DW was inhibited by 53.57%, stem DW by 52.56% and leaf DW by 43.10% (Table 1). At 100 mM NaCl, reductions for root DW, stem DW and leaf DW were 42.86%, 44.87% and 35.34%, respectively. At 100 mM NaCl no significant differences were noted between the pretreatments, and mean inhibitions were only 19.29% (Figure 2). At 200 mM NaCl, the whole plant DW declined more in GA$_3$-pre-treated plants (30.70%) than in the other pretreatments (average inhibition 25.04%; $p < 0.05$). The highest values for leaf dry weight at 100 mM were in GA$_3$ plants (1.07 g plant$^{-1}$), while in the other pretreatments no significant differences were detected (average inhibition of 18.39%) (Table 1). At 200 mM, leaf dry weight was inhibited by 20.90% in pretreated plants ($p < 0.05$). Salinity inhibited stem growth least in plants pretreated with IAA (14.10 and 16.67% for 100 and 200 mM

NaCl, respectively), while in root dry weight the least inhibition was detected in plants pretreated with SA and Kinetin (average inhibition of 20.53 and 25.00% for 100 and 200 mM NaCl, respectively). Salinity significantly reduced ($p < 0.05$) the leaf water content (LRWC) in a concentration-dependent fashion (Table 1). Plants pretreated with Kinetin and IAA had greater LRWC values than no-PGR plants at both levels of NaCl (100 and 200 mM NaCl).

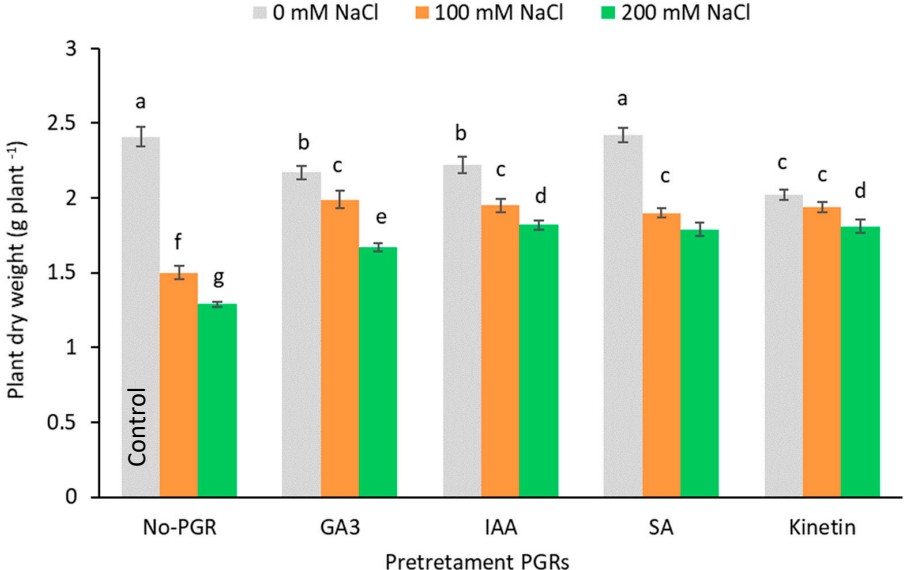

**Figure 2.** Effect of salt stress on plant dry weight of *Olea europaea* plants pretreated with plant growth regulators (PGRs). Data are expressed as mean ± SE (n = 6). Means followed by the same letter are not significantly different according to the LSD test ($p < 0.05$).

**Table 1.** Effect of salt stress on root dry weight, stem dry weight, leaf dry weight, and leaf relative water content (LRWC) of *Olea europaea* plants pretreated with plant growth regulators (PGRs). Data are expressed as mean ± SE (n = 6).

| PGRs without NaCl | Root Dry Weight (g plant$^{-1}$) | Stem Dry Weight (g plant$^{-1}$) | Leaf Dry Weight (g plant$^{-1}$) | LRWC (%) |
|---|---|---|---|---|
| Control: 0 mM NaCl (no-PGR) | 0.56 ± 0.023 [a] | 0.78 ± 0.026 [a] | 1.16 ± 0.030 [ab] | 93.37 ± 0.351 [a] |
| GA₃ + 0 mM | 0.43 ± 0.013 [cd] | 0.66 ± 0.025 [b] | 1.08 ± 0.048 [bc] | 83.21 ± 0.525 [e] |
| IAA + 0 mM | 0.36 ± 0.012 [e] | 0.55 ± 0.019 [cd] | 1.05 ± 0.042 [c] | 93.26 ± 0.350 [a] |
| SA + 0 mM | 0.57 ± 0.021 [a] | 0.68 ± 0.031 [b] | 1.17 ± 0.033 [a] | 85.46 ± 0.453 [cd] |
| Kinetin + 0 mM | 0.50 ± 0.014 [b] | 0.51 ± 0.008 [c] | 1.00 ± 0.024 [cd] | 88.76 ± 0.701 [b] |
| **PGRs with 100 mM NaCl** | **Root dry weight (g plant$^{-1}$)** | **Stem dry weight (g plant$^{-1}$)** | **Leaf dry weight (g plant$^{-1}$)** | **LRWC (%)** |
| 100 mM (no-PGR) | 0.32 ± 0.007 [ef] | 0.43 ± 0.015 [d] | 0.75 ± 0.030 [f] | 82.30 ± 0.276 [ef] |
| GA₃ + 100 mM | 0.40 ± 0.014 [d] | 0.52 ± 0.022 [c] | 1.07 ± 0.041 [c] | 82.13 ± 0.471 [ef] |
| IAA + 100 mM | 0.34 ± 0.013 [ef] | 0.67 ± 0.030 [b] | 0.93 ± 0.033 [de] | 86.30 ± 0.653 [c] |
| SA + 100 mM | 0.43 ± 0.015 [cd] | 0.43 ± 0.015 [d] | 0.96 ± 0.037 [de] | 82.49 ± 0.611 [ef] |
| Kinetin + 100 mM | 0.46 ± 0.011 [c] | 0.52 ± 0.022 [c] | 0.95 ± 0.029 [de] | 84.60 ± 0.456 [d] |
| **PGRs with 200 mM NaCl** | **Root dry weight (g plant$^{-1}$)** | **Stem dry weight (g plant$^{-1}$)** | **Leaf dry weight (g plant$^{-1}$)** | **LRWC (%)** |
| 200 mM (no-PGR) | 0.26 ± 0.008 [g] | 0.37 ± 0.015 [e] | 0.66 ± 0.016 [g] | 77.84 ± 0.471 [h] |
| GA₃ + 200 mM | 0.26 ± 0.010 [g] | 0.51 ± 0.021 [c] | 0.89 ± 0.019 [e] | 78.75 ± 0.565 [h] |
| IAA + 200 mM | 0.26 ± 0.010 [g] | 0.65 ± 0.023 [b] | 0.90 ± 0.036 [e] | 81.41 ± 0.210 [fg] |
| SA + 200 mM | 0.36 ± 0.011 [e] | 0.49 ± 0.021 [c] | 0.94 ± 0.040 [de] | 77.80 ± 0.524 [h] |
| Kinetin + 200 mM | 0.34 ± 0.012 [e] | 0.52 ± 0.010 [c] | 0.94 ± 0.020 [de] | 79.25 ± 0.436 [g] |

Means followed by the same letter within the same column are not significantly different according to the LSD test ($p < 0.05$).

### 3.2. Effects of the Application of Plant Growth Regulators (PGRs) on Tissue Mineral Concentration

Ion ($Na^+$ and $K^+$) concentration was determined in the roots and leaves from plants under different treatments (Tables 2 and 3). Pretreatments did not significantly affect leaf $Na^+$ concentration without salt. Under saline conditions, however, the $Na^+$ concentration increased in both pretreated and non-pretreated PGR plants, being more noticeable in roots than in leaves. Nevertheless, PGRs with salt accumulated less $Na^+$ in leaves and roots than non-pretreated plants. Pretreatments with IAA and SA gave the lowest $Na^+$ concentrations at both NaCl levels (IAA+200 mM = 472.06 and SA+200 mM = 414.96 mmol kg$^{-1}$ DW), whereas with these pretreatments $Na^+$ accumulation occurred in roots. Furthermore, at 200 mM NaCl, plants pretreated with SA had the lowest net rates of $Na^+$ translocation ($J_{s\text{-}Na}$; Figure 3A) from roots to leaves (30.94 mmol $Na^+$ kg$^{-1}$ root DW day$^{-1}$), followed by plants pretreated with IAA (39.48 mmol $Na^+$ kg$^{-1}$ root DW day$^{-1}$). Therefore, compared to plants with no-PGR+200 mM NaCl and those pretreated with SA and IAA, $J_{s\text{-}Na}$ was reduced by 40.76% and 24.41%, respectively. However, in plants with GA$_3$+200 mM, Kinetin+200 mM, and no-PGR+200 mM, no differences were noted for $J_{s\text{-}Na}$. At 100 mM NaCl, the SA pretreatment had the lowest $J_{s\text{-}Na}$ (27.39 mmol $Na^+$ kg$^{-1}$ root DW day$^{-1}$), followed by the IAA (23.26 mmol $Na^+$ kg$^{-1}$ root DW day$^{-1}$) and Kinetin pretreatments (35.39 mmol $Na^+$ kg$^{-1}$ root DW day$^{-1}$). This led to a drop in $J_{s\text{-}Na}$ of 49.53% with SA pretreatment, 40.57% with IAA, and 23.26% with Kinetin compared to no-PGR+100 mM NaCl plants. Therefore, $Na^+$ transport from roots to leaves was determined by the pretreatment and by the level of added salt. In light of these results, the treatments can be divided into three groups. The first is composed of plants pretreated with GA$_3$ whose $J_{s\text{-}Na}$ was similar to that of the no-PGR plants. The second group includes Kinetin-pretreated plants that had less $Na^+$ translocation at 100 mM NaCl than no-PGR plants, but greater translocation than plants pretreated with IAA and SA. The final group includes the pretreatments with IAA and SA that had the lowest values for $J_{s\text{-}Na}$ at both NaCl concentrations.

**Table 2.** Effect of salt stress on $Na^+$ and $K^+$ concentrations, and $K^+/Na^+$ ratio in leaf of *Olea europaea* plants pretreated with plant growth regulators (PGRs). Data are expressed as mean $\pm$ SE (n=4).

| | Leaf | | |
|---|---|---|---|
| **PGRs without NaCl** | **$Na^+$ (mmol kg$^{-1}$ DW)** | **$K^+$ (mmol kg$^{-1}$ DW)** | **$K^+/Na^+$ Ratio** |
| Control: 0 mM NaCl (no-PGR) | 66.96 ± 0.50 [gh] | 357.38 ± 1.56 [a] | 4.900 ± 0.023 [c] |
| GA$_3$ + 0 mM | 53.37 ± 3.86 [gh] | 299.03 ± 4.31 [c] | 5.603 ± 0.081 [b] |
| IAA + 0 mM | 76.31 ± 2.34 [gh] | 305.12 ± 1.08 [c] | 3.999 ± 0.014 [d] |
| SA + 0 mM | 98.01 ± 0.17 [g] | 331.81 ± 0.71 [b] | 3.386 ± 0.007 [e] |
| Kinetin + 0 mM | 47.25 ± 2.84 [h] | 289.14 ± 2.86 [d] | 6.120 ± 0.061 [a] |
| **PGRs with 100 mM NaCl** | **$Na^+$ (mmol kg$^{-1}$ DW)** | **$K^+$ (mmol kg$^{-1}$ DW)** | **$K^+/Na^+$ ratio** |
| 100 mM NaCl (no-PGR) | 610.13 ± 28.70 [b] | 247.15 ± 1.51 [fg] | 0.405 ± 0.002 [ij] |
| GA$_3$ + 100 mM | 490.22 ± 1.11 [d] | 224.25 ± 4.19 [h] | 0.457 ± 0.009 [i] |
| IAA + 100 mM | 347.57 ± 17.26 [f] | 265.86 ± 5.42 [e] | 0.765 ± 0.015 [f] |
| SA + 100 mM | 334.68 ± 16.64 [f] | 264.06 ± 3.04 [e] | 0.789 ± 0.009 [f] |
| Kinetin + 100 mM | 457.37 ± 23.00 [de] | 246.81 ± 2.17 [g] | 0.540 ± 0.005 [h] |
| **PGRs with 200 mM NaCl** | **$Na^+$ (mmol kg$^{-1}$ DW)** | **$K^+$ (mmol kg$^{-1}$ DW)** | **$K^+/Na^+$ ratio** |
| 200 mM NaCl (no-PGR) | 693.93 ± 12.25 [a] | 252.23 ± 3.42 [fg] | 0.364 ± 0.005 [j] |
| GA$_3$ + 200 mM | 552.14 ± 31.25 [c] | 251.55 ± 4.84 [fg] | 0.456 ± 0.009 [i] |
| IAA + 200 mM | 472.06 ± 2.18 [d] | 302.96 ± 2.83 [c] | 0.642 ± 0.006 [g] |
| SA + 200 mM | 414.96 ± 23.37 [e] | 267.24 ± 2.47 [e] | 0.644 ± 0.006 [g] |
| Kinetin + 200 mM | 593.00 ± 32.77 [bc] | 256.08 ± 2.34 [f] | 0.432 ± 0.004 [ij] |

Means followed by the same letter within the same column are not significantly different according to the LSD test ($p < 0.05$).

**Table 3.** Effect of salt stress on $Na^+$ and $K^+$ concentrations, and $K^+/Na^+$ ratio in root of *Olea europaea* plants pretreated with plant growth regulators (PGRs). Data are expressed as mean $\pm$ SE (n=4).

| PGRs without NaCl | Root | | |
|---|---|---|---|
| | $Na^+$ (mmol kg$^{-1}$ DW) | $K^+$ (mmol kg$^{-1}$ DW) | $K^+/Na^+$ Ratio |
| Control: 0 mM NaCl (no-PGR) | 122.62 $\pm$ 0.71 [k] | 856.44 $\pm$ 2.96 [a] | 6.984 $\pm$ 0.024 [b] |
| GA$_3$ + 0 mM | 159.88 $\pm$ 0.69 [j] | 603.58 $\pm$ 1.90 [e] | 3.775 $\pm$ 0.012 [e] |
| IAA + 0 mM | 120.54 $\pm$ 0.91 [k] | 835.34 $\pm$ 0.73 [b] | 6.930 $\pm$ 0.006 [c] |
| SA + 0 mM | 159.13 $\pm$ 0.48 [j] | 686.18 $\pm$ 0.59 [d] | 4.312 $\pm$ 0.004 [d] |
| Kinetin + 0 mM | 97.28 $\pm$ 0.17 [l] | 750.89 $\pm$ 1.15 [c] | 7.719 $\pm$ 0.012 [a] |
| **PGRs with 100 mM NaCl** | $Na^+$ (mmol kg$^{-1}$ DW) | $K^+$ (mmol kg$^{-1}$ DW) | $K^+/Na^+$ ratio |
| 100 mM NaCl (no-PGR) | 1790.29 $\pm$ 8.15 [b] | 367.19 $\pm$ 1.21 [g] | 0.205 $\pm$ 0.002 [g] |
| GA$_3$ + 100 mM | 1187.19 $\pm$ 2.55 [h] | 309.95 $\pm$ 1.24 [i] | 0.261 $\pm$ 0.001 [f] |
| IAA + 100 mM | 1594.39 $\pm$ 1.21 [e] | 397.57 $\pm$ 2.90 [f] | 0.249 $\pm$ 0.001 [f] |
| SA + 100 mM | 1232.79 $\pm$ 6.06 [g] | 210.26 $\pm$ 1.63 [l] | 0.177 $\pm$ 0.001 [h] |
| Kinetin + 100 mM | 1144.83 $\pm$ 0.37 [i] | 170.35 $\pm$ 2.27 [m] | 0.149 $\pm$ 0.002 [i] |
| **PGRs with 200 mM NaCl** | $Na^+$ (mmol kg$^{-1}$ DW) | $K^+$ (mmol kg$^{-1}$ DW) | $K^+/Na^+$ ratio |
| 200 mM NaCl (no-PGR) | 1899.94 $\pm$ 1.51 [a] | 267.02 $\pm$ 3.64 [j] | 0.141 $\pm$ 0.002 [i] |
| GA$_3$ + 200 mM | 1392.42 $\pm$ 2.43 [f] | 124.27 $\pm$ 1.17 [n] | 0.089 $\pm$ 0.001 [j] |
| IAA + 200 mM | 1778.69 $\pm$ 3.61 [c] | 323.92 $\pm$ 4.27 [h] | 0.182 $\pm$ 0.002 [h] |
| SA + 200 mM | 1658.42 $\pm$ 2.57 [d] | 243.37 $\pm$ 2.99 [k] | 0.147 $\pm$ 0.002 [i] |
| Kinetin + 200 mM | 1592.74 $\pm$ 3.21 [e] | 162.79 $\pm$ 2.37 [m] | 0.102 $\pm$ 0.001 [j] |

Means followed by the same letter within the same column are not significantly different according to the LSD test ($p < 0.05$).

In plants grown with no NaCl, the $K^+$ concentration of leaves had the lowest values in plants pretreated with Kinetin, in which the highest $K^+/Na^+$ ratios were detected. In general, salinity significantly decreased ($p < 0.05$) the $K^+$ concentration of both roots and leaves (Tables 2 and 3). In leaves, plants pretreated with IAA had the smallest fall in $K^+$ concentration at both NaCl levels (18.79 and 7.46% at 100 and 200 mM NaCl, respectively) compared to the control plants. Interestingly, in the SA pretreatment the $K^+$ concentration of plants grown at 200 mM NaCl only decreased by 18.37% compared to control plants. In addition, in the IAA and SA pretreatments, plants had greater $K^+/Na^+$ leaf ratios at both salt levels (100 and 200 mM NaCl). By contrast, in GA$_3$, Kinetin, and no-PGR plants, at 200 mM NaCl, no significant differences ($p < 0.05$) in leaf $K^+$ concentration were detected, whereas at 100 mM NaCl GA$_3$ plants had the lowest $K^+$ concentration (224.25 mmol kg$^{-1}$ DW). Under salt stress, the net rates of $K^+$ translocation ($J_{s\text{-}K}$) from roots to leaves were higher in plants pretreated with GA$_3$ and Kinetin (Figure 3B). In roots at 100 and 200 mM NaCl, the smallest drop in $K^+$ concentration was detected in plants pretreated with IAA (53.58 and 62.18%, respectively), in which $K^+/Na^+$ values were highest (Table 3).

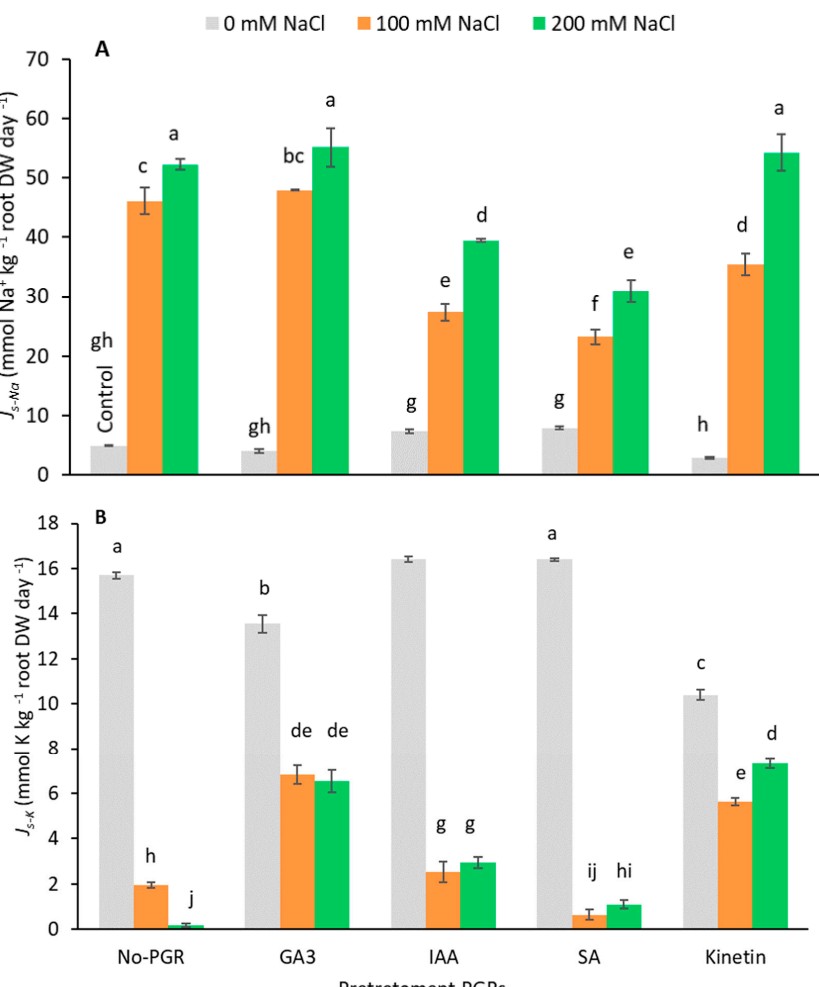

**Figure 3.** Effect of salt stress on net translocation rates of Na$^+$ (**A**) and K$^+$ (**B**) from root to leaves of *Olea europaea* plants pretreated with plant growth regulators (PGRs). Data are expressed as mean ± SE (n = 4). Means followed by the same letter are not significantly different according to the LSD test ($p < 0.05$).

### 3.3. Effects of the Application Plant Growth Regulators (PGRs) on the Chlorophyll Content

Pigment analysis showed that the pretreatments caused different responses to stress (Figure 4). Under salt-stress, Chl($a+b$) and carotenoid content decreased compared to the control in no-PGR plants and in plants pretreated with GA$_3$ (Figure 4A,C). Conversely, in plants pretreated with IAA, SA, and Kinetin, the levels of these pigments were higher than in control plants at all tested salt levels, the highest value being found in plants pretreated with Kinetin. In general, the chlorophyll content decreased relative to salt levels. In no-PGR plants, the chlorophyll content decreased by 22.53% at 100 mM and by 41.90% at 200 mM NaCl compared to the control (Figure 4A). However, plants with Kinetin+200 mM NaCl had a chlorophyll content that was 110.91% higher than no-PGR+200 mM plants, whereas in plants with IAA+200 mM and SA+200 mM the chlorophyll content was 76.97% higher than in no-PGR plants+200 mM NaCl. At 100 mM NaCl, compared to no-PGR+100 mM NaCl plants, the increases in chlorophyll content were 73.18%, 59.09%, and 44.41% in plants pretreated with Kinetin, SA, and IAA, respectively. Compared to the control, in no-PGR plants, the carotenoid content decreased by 13.47% and 31.37% at 100 mM and 200 mM NaCl, respectively (Figure 4C). In plants pretreated with IAA, SA, and Kinetin, the carotenoid content was higher than in no-PGR plants. At 200 mM, the increases compared to no-PGR+200 mM were 117.94%, 87.80% and 83.97% for plants pretreated with IAA, SA and IAA, respectively. The Chl$a$/Chl$b$ ratio (Figure 4B) in no-PGR plants decreased at

200 mM NaCl compared to the control. Therefore, Chl*a* was found to be more sensitive to salinity than Chl*b*. In plants pretreated with IAA, SA, and Kinetin, the Chl*a*/Chl*b* ratio was higher than in the control plants for all salt levels, whereas in the pretreatment with GA$_3$ it was lower at both 100 and 200 mM NaCl. However, with GA$_3$+200 mM, the Chl*a*/Chl*b* ratio was 5.35% higher than in no-PGR plants+200 mM NaCl.

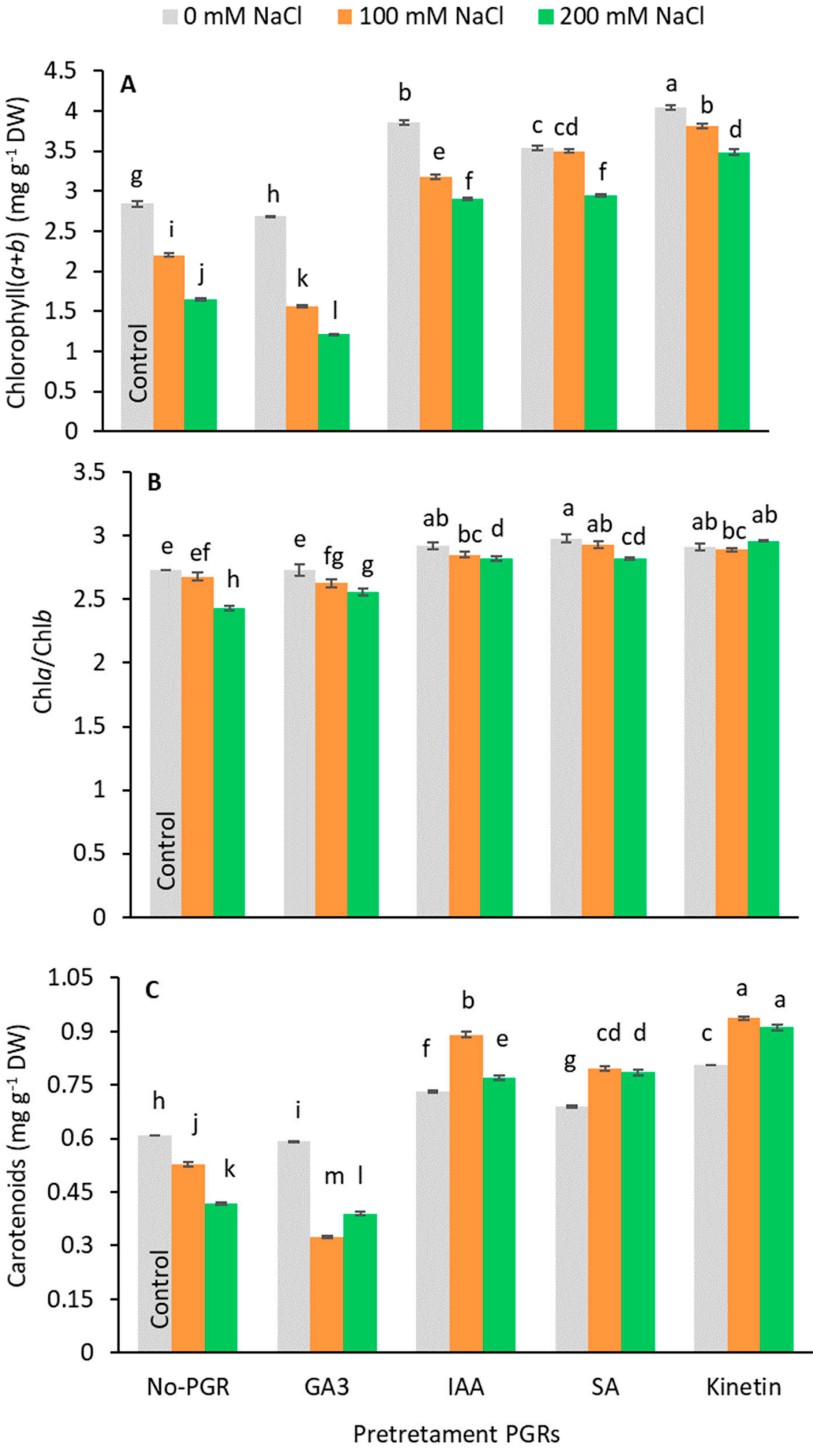

**Figure 4.** Effect of salt stress on chlorophyll total (**A**), ratio Chlorophyll *a*/Chlorophyll *b* (**B**), and carotenoids (**C**) in leaf tissue of *Olea europaea* plants pretreated with plant growth regulators (PGRs). Data are expressed as mean ± SE (n = 4). Means followed by the same letter are not significantly different according to the LSD test ($p < 0.05$).

### 3.4. Effects of the Application of Plant Growth Regulators (PGRs) on Leaf Organic Solutes and Starch

The pretreatments significantly affected ($p < 0.05$) proline, fructose, glucose, inositol, mannitol, sucrose, and starch contents (Tables 4 and 5). At 100 mM NaCl, the leaf proline concentration of plants increased significantly compared to the control in plants not pretreated with PGR (12.28%) and plants pretreated with IAA (21.18%), SA (19.26%) and Kinetin (19.32%) (Table 4). According to the LSD test, no differences were observed between the proline levels attained in no-PGR plants and in plants pretreated with IAA, SA, and Kinetin ($p < 0.05$). At 200 mM NaCl, only plants pretreated with IAA and SA accumulated proline (15.65% on average), while in GA$_3$ plants proline concentration decreased at both 100 and 200 mM NaCl (15.66% on average). However, in plants with no NaCl, the highest proline values were observed in plants with GA$_3$ (GA$_3$+0 mM NaCl).

**Table 4.** Effect of salt stress on proline, sucrose, and starch concentrations in leaf *Olea europaea* plants pretreated with plant growth regulators (PGRs). Data are expressed as mean $\pm$ SE (n = 4).

| PGRs without NaCl | Proline ($\mu$g g$^{-1}$ DW) | Sucrose (mg g$^{-1}$ DW) | Starch (mg g$^{-1}$ DW) |
|---|---|---|---|
| Control: 0 mM NaCl (no PGR) | 167.60 $\pm$ 4.64 [c] | 13.40 $\pm$ 0.09 [ab] | 16.11 $\pm$ 0.63 [cd] |
| GA$_3$ + 0 mM | 183.63 $\pm$ 4.24 [ab] | 12.41 $\pm$ 0.12 [d] | 16.47 $\pm$ 0.14 [cd] |
| IAA + 0 mM | 137.29 $\pm$ 3.05 [d] | 13.47 $\pm$ 0.26 [ab] | 2.42 $\pm$ 0.51 [f] |
| SA + 0 mM | 159.26 $\pm$ 2.57 [c] | 11.72 $\pm$ 0.19 [e] | 21.68 $\pm$ 1.92 [b] |
| Kinetin + 0 mM | 163.32 $\pm$ 1.76 [c] | 13.77 $\pm$ 0.14 [a] | 15.71 $\pm$ 0.59 [d] |
| **PGRs with 100 mM NaCl** | **Proline ($\mu$g g$^{-1}$ DW)** | **Sucrose (mg g$^{-1}$ DW)** | **Starch (mg g$^{-1}$ DW)** |
| 100 mM NaCl (no PGR) | 188.19 $\pm$ 2.51 [ab] | 13.66 $\pm$ 0.24 [ab] | 16.28 $\pm$ 0.78 [cd] |
| GA$_3$ + 100 mM | 133.90 $\pm$ 0.55 [d] | 13.46 $\pm$ 0.07 [ab] | 22.31 $\pm$ 0.78 [b] |
| IAA + 100 mM | 203.10 $\pm$ 3.17 [a] | 11.17 $\pm$ 0.20 [e] | 2.89 $\pm$ 0.35 [f] |
| SA + 100 mM | 199.94 $\pm$ 4.20 [a] | 11.60 $\pm$ 0.25 [e] | 15.63 $\pm$ 0.62 [d] |
| Kinetin + 100 mM | 199.98 $\pm$ 4.11 [a] | 12.81 $\pm$ 0.27 [cd] | 30.44 $\pm$ 0.57 [a] |
| **PGRs with 200 mM NaCl** | **Proline ($\mu$g g$^{-1}$ DW)** | **Sucrose (mg g$^{-1}$ DW)** | **Starch (mg g$^{-1}$ DW)** |
| 200 mM NaCl (no PGR) | 171.02 $\pm$ 1.62 [c] | 6.90 $\pm$ 0.16 [h] | 5.79 $\pm$ 0.11 [e] |
| GA$_3$ + 200 mM | 158.25 $\pm$ 2.84 [d] | 9.18 $\pm$ 0.22 [f] | 1.05 $\pm$ 0.05 [f] |
| IAA + 200 mM | 197.19 $\pm$ 1.89 [a] | 8.36 $\pm$ 0.14 [g] | 6.31 $\pm$ 0.26 [e] |
| SA + 200 mM | 190.51 $\pm$ 3.62 [ab] | 8.13 $\pm$ 0.16 [g] | 17.97 $\pm$ 0.19 [c] |
| Kinetin + 200 mM | 167.85 $\pm$ 3.24 [c] | 6.79 $\pm$ 0.32 [h] | 2.72 $\pm$ 0.63 [f] |

Means followed by the same letter within the same column are not significantly different according to the LSD test ($p < 0.05$).

The leaves of plants grown without salt and pretreated with IAA had the highest levels of fructose, inositol, and sucrose but the lowest starch levels (Tables 4 and 5). By contrast, plants pretreated with SA had the highest starch levels and the lowest sucrose, mannitol, inositol, and glucose levels. Under salt stress, no-PGR plants accumulated glucose at both salt levels and mannitol at 100 mM NaCl. Plants pretreated with GA$_3$ only accumulated mannitol (100 and 200 mM NaCl), whereas plants pretreated with IAA, SA, and Kinetin accumulated fructose (100 and 200 mM NaCl), mannitol (100 and 200 mM NaCl), and glucose (only 200 mM NaCl). Plants pretreated with IAA had the highest increases compared to the control in fructose at 100 mM (40.76%) and 200 mM NaCl (59.91%), glucose at 200 mM (117.01%), and mannitol at 200 mM (58.70%).

In no-PGR and salt-stressed (100 and 200 mM NaCl) plants, glucose accumulation was favored over mannitol accumulation, as revealed by the significant decrease in the mannitol/glucose ratio compared to the control (27.71% on average) (Figure 5). Conversely, in plants pretreated with GA$_3$, IAA, SA, and Kinetin grown at 100 mM NaCl, the mannitol/glucose ratio rose, with the greatest increase being observed in plants pretreated with

IAA (119.51%), followed by plants pretreated with SA, Kinetin (100%), and GA$_3$ (57.24%). However, glucose accumulation at 200 mM NaCl led to a fall in the mannitol/glucose ratio in plants pretreated with IAA, SA, and Kinetin. Compared to the control, the starch concentration (Table 4) only increased in plants grown at 100 mM NaCl and pretreated with GA$_3$ (22.31 mg g$^{-1}$ DW) and Kinetin (30.44 mg g$^{-1}$ DW).

**Table 5.** Effect of salt stress on fructose, glucose, inositol, and mannitol concentrations in leaf *Olea europaea* plants pretreated with plant growth regulators (PGRs). Data are expressed as mean ± SE (n = 4).

| PGRs without NaCl | Fructose (mg g$^{-1}$ DW) | Glucose (mg g$^{-1}$ DW) | Inositol (mg g$^{-1}$ DW) | Mannitol (mg g$^{-1}$ DW) |
|---|---|---|---|---|
| Control: 0 mM NaCl (no PGR) | 14.99 ± 0.19 [e] | 23.10 ± 0.35 [e] | 1.59 ± 0.046 [a] | 22.52 ± 1.41 [d] |
| GA$_3$ + 0 mM | 21.36 ± 0.12 [b] | 14.96 ± 0.91 [g] | 0.86 ± 0.054 [d] | 17.32 ± 1.23 [e] |
| IAA + 0 mM | 22.46 ± 0.44 [b] | 30.77 ± 0.73 [d] | 1.66 ± 0.052 [a] | 17.79 ± 0.34 [e] |
| SA + 0 mM | 19.36 ± 0.37 [c] | 15.60 ± 0.90 [g] | 0.76 ± 0.034 [e] | 14.61 ± 0.87 [f] |
| Kinetin + 0 mM | 17.49 ± 0.22 [d] | 37.12 ± 1.21 [c] | 1.25 ± 0.036 [b] | 17.54 ± 0.76 [e] |
| **PGRs with 100 mM NaCl** | **Fructose (mg g$^{-1}$ DW)** | **Glucose (mg g$^{-1}$ DW)** | **Inositol (mg g$^{-1}$ DW)** | **Mannitol (mg g$^{-1}$ DW)** |
| 100 mM NaCl (no PGR) | 14.60 ± 0.10 [e] | 45.24 ± 2.80 [b] | 0.90 ± 0.013 [d] | 29.79 ± 0.57 [c] |
| GA$_3$ + 100 mM | 14.90 ± 0.13 [e] | 19.34 ± 0.82 [f] | 0.87 ± 0.031 [d] | 29.64 ± 1.62 [c] |
| IAA + 100 mM | 21.10 ± 0.06 [b] | 14.99 ± 1.48 [g] | 1.01 ± 0.060 [c] | 33.07 ± 1.11 [b] |
| SA + 100 mM | 17.49 ± 0.38 [d] | 15.58 ± 0.60 [g] | 0.77 ± 0.041 [e] | 30.30 ± 1.21 [c] |
| Kinetin + 100 mM | 21.49 ± 0.06 [b] | 17.32 ± 0.87 [fg] | 0.62 ± 0.025 [g] | 33.94 ± 0.83 [b] |
| **PGRs with 200 mM NaCl** | **Fructose (mg g$^{-1}$ DW)** | **Glucose (mg g$^{-1}$ DW)** | **Inositol (mg g$^{-1}$ DW)** | **Mannitol (mg g$^{-1}$ DW)** |
| 200 mM NaCl (no PGR) | 13.95 ± 0.18 [ef] | 31.85 ± 0.27 [d] | 0.50 ± 0.020 [h] | 23.92 ± 0.27 [d] |
| GA$_3$ + 200 mM | 13.04 ± 0.12 [f] | 23.92 ± 0.90 [e] | 0.69 ± 0.015 [f] | 33.71 ± 1.66 [b] |
| IAA + 200 mM | 23.97 ± 0.42 [a] | 50.13 ± 0.77 [a] | 0.40 ± 0.021 [h] | 35.74 ± 1.21 [a] |
| SA + 200 mM | 16.84 ± 0.15 [d] | 36.57 ± 1.55 [c] | 0.64 ± 0.040 [g] | 30.53 ± 0.37 [c] |
| Kinetin + 200 mM | 16.45 ± 0.35 [d] | 46.77 ± 2.10 [b] | 0.48 ± 0.019 [h] | 37.95 ± 1.12 [a] |

Means followed by the same letter within the same column are not significantly different according to the LSD test (*p* < 0.05).

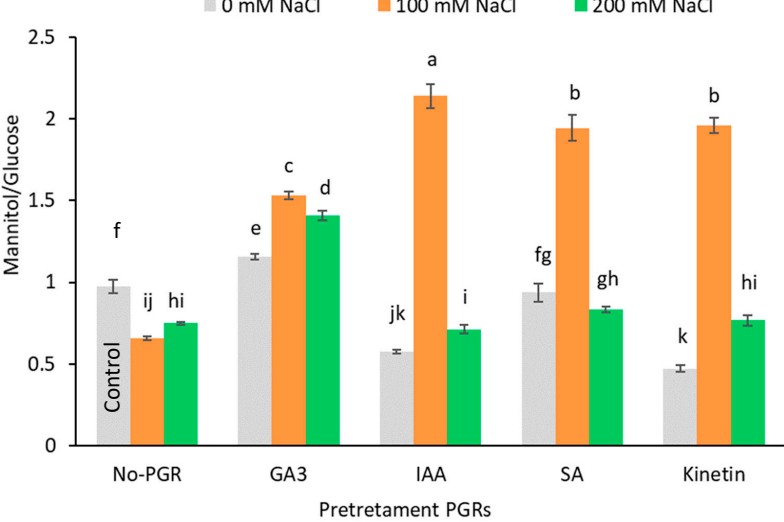

**Figure 5.** Effect of salt stress on ratio Manitol/Glucose in leaf of *Olea europaea* plants pretreated with plant growth regulators (PGRs). Data are expressed as mean ± SE (n = 4). Means followed by the same letter are not significantly different according to the LSD test (*p* < 0.05).

### 3.5. Effects of the Application of Plant Growth Regulators (PGRs) on Free Polyamine Content and Ethylene production

The endogenous Put, Spd, and Spm content were analyzed. Putrescine was not detected in any plant. In plants grown with no NaCl, the Spd content increased in those pretreated with $GA_3$ (19.87%) and Kinetin (79.80%) (Figure 6A). Under saline stress, the Spd content only increased compared to the control at 200 mM NaCl in no-PGR plants (29.03%) and plants pretreated with IAA (56.18%) and SA (31.95%). By contrast, in plants pretreated with $GA_3$ and Kinetin, the Spd content dropped as salinity increased. The Spm content (Figure 6B) increased with $GA_3$+100 mM (23.79%), IAA+0 mM (75.81%), IAA+100 mM (80.91%), IAA+200 mM (56.18%), SA+0 mM (142.69%), SA+100 mM (147.56%), SA+200 mM (154.16%), and Kinetin+200 mM (75.83%), with the highest values being observed in plants pretreated with SA, in which no differences were detected between the three levels of salts assessed.

Compared to the control, ethylene production (Figure 7) fell in all pretreatments at 100 mM NaCl but increased at 200 mM in IAA-pretreated plants and reached its highest levels (4.44 nmol $g^{-1}$ FW $h^{-1}$) in no-PGR plants.

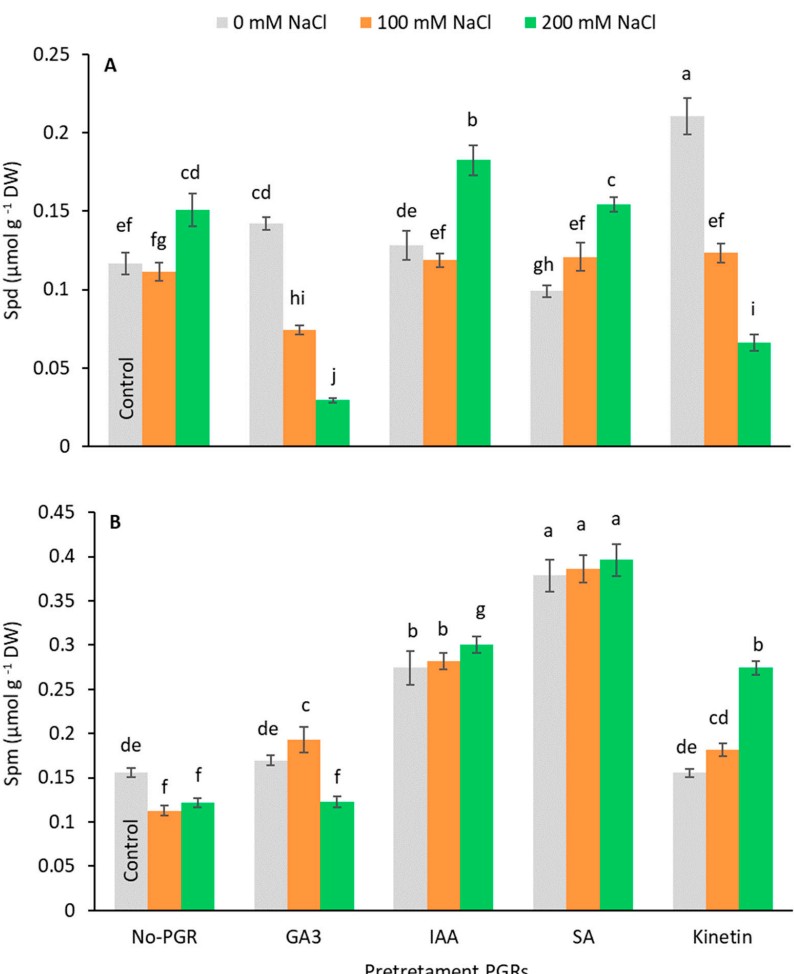

**Figure 6.** Effect of salt stress on Spd (**A**) and Spm (**B**) contents in leaf of *Olea europaea* plants pretreated with plant growth regulators (PGRs). Data are expressed as mean $\pm$ SE (n = 4). Means followed by the same letter are not significantly different according to the LSD test ($p < 0.05$).

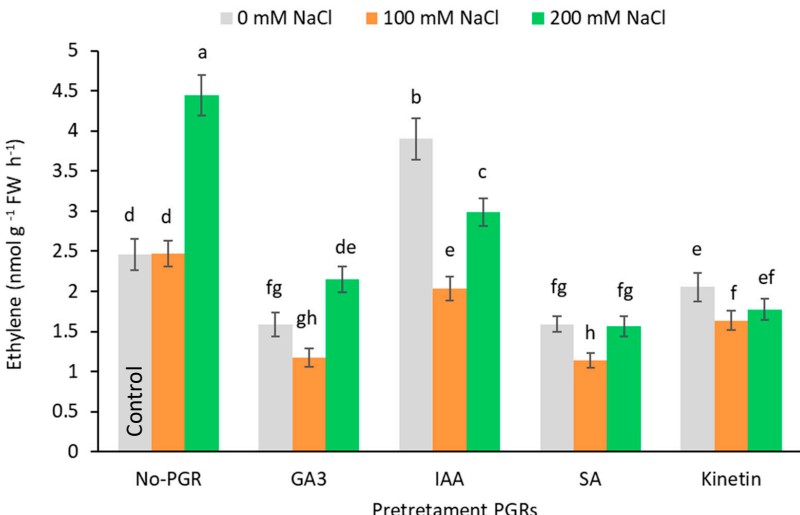

**Figure 7.** Effect of salt stress on ethylene production by leaf of *Olea europaea* plants pretreated with plant growth regulators (PGRs). Data are expressed as mean $\pm$ SE (n = 6). Means followed by the same letter are not significantly different according to the LSD test ($p < 0.05$).

### 3.6. Multivariable Analysis

The most striking results include the significant negative correlation found between whole plant dry weight and leaf $Na^+$ ($-0.8690$), root $Na^+$ ($-0.8455$), $J_{s-Na}$ ($-0.7999$), and mannitol ($-0.5269$) and, by contrast, the positive correlation found between plant dry weight and $J_{s-K}$ (0.8350), leaf $K^+$ (0.7212), root $K^+$ (0.7182), root $K^+/Na^+$ (0.7052), leaf $K^+/Na^+$ (0.6949), sucrose (0.5544), and chlorophyll $a/b$ (0.5468) (Table 6). Additionally, carotenoids (0.8425) and Spm (0.6689) correlated with chlorophyll $a/b$, while mannitol was highly correlated with root $K^+$ ($-0.8320$), leaf $K^+/Na^+$ ($-0.8294$), root $Na^+$ (0.8182), root $K^+/Na^+$ ($-0.8107$), $J_{s-Na}$ (0.7788), leaf Na+ (0.7630), and $J_{s-K}$ ($-0.7032$).

**Table 6.** Simple correlation coefficient (Pearson method) among all parameters studied in saline and non-saline conditions in *Olea europea* ($p \leq 0.05$ *; $p \leq 0.01$ **). Abbreviations: PDW (plant dry weight); RDW (root dry weight); SDW (stem dry weight); LDW (leaf dry weight); LRWC (leaf relative water content); $J_{s-Na}$ (net translocation of $Na^+$); $J_{s-K}$ (net translocation of $K^+$); Chl (chlorophyll $a+b$); Chl$a/b$ (chlorophyll $a/b$); Car (carotenoids); Spd (spermidine); Spm (spermine).

| | PDW | RDW | SDW | LDW | LRWC | Leaf $Na^+$ | Leaf $K^+$ |
|---|---|---|---|---|---|---|---|
| PDW | 1 | | | | | | |
| RDW | 0.8092 ** | 1 | | | | | |
| SDW | 0.7951 ** | 0.4840 | 1 | | | | |
| LDW | 0.9264 ** | 0.6428 ** | 0.6175 * | 1 | | | |
| LRWC | 0.7559 ** | 0.6077 * | 0.5570 * | 0.7245 ** | 1 | | |
| Leaf $Na^+$ | −0.8690 ** | −0.7438 ** | −0.6246 * | −0.8158 ** | −0.7737 ** | 1 | |
| Leaf $K^+$ | 0.7212 ** | 0.5228 * | 0.7212 ** | 0.6091 * | 0.6070 * | −0.7841 ** | 1 |
| Leaf $K^+/Na^+$ | 0.6949 ** | 0.6545 ** | 0.5045 | 0.6365 * | 0.6998 ** | −0.9093 ** | 0.7239 ** |
| Root $Na^+$ | −0.8455 ** | −0.7809 ** | −0.5252 * | −0.8300 ** | −0.7650 ** | 0.9304 ** | −0.6926 ** |
| Root $K^+$ | 0.7182 ** | 0.6002 * | 0.5595 * | 0.6955 ** | 0.8494 ** | −0.8569 ** | 0.7818 ** |
| Root $K^+/Na^+$ | 0.7052 ** | 0.6388 * | 0.4574 | 0.8339 ** | 0.8339 ** | −0.9199 ** | 0.7354 ** |
| $J_{s-Na}$ | −0.7999 ** | −0.7435 ** | −0.5577 * | −0.7174 ** | −0.7563 ** | 0.9764 ** | −0.7861 ** |
| $J_{s-K}$ | 0.8350 ** | 0.6437 ** | 0.6306* | 0.8501 ** | 0.7150 ** | −0.7894 ** | 0.7049 ** |
| Chl | 0.5039 | 0.5230 * | 0.2466 | 0.4662 | 0.5401 * | −0.5616 * | 0.4609 |
| Chl$a/b$ | 0.5468 * | 0.4873 | 0.3065 | 0.5326 * | 0.4120 | −0.4890 | 0.4066 |
| Car | 0.2412 | 0.2321 | 0.1985 | 0.1669 | 0.2457 | −0.2362 | 0.2293 |
| Proline | −0.2345 | −0.1155 | 0.0231 | −0.4396 | −0.1977 | 0.1183 | 0.0663 |
| Fructose | 0.3904 | 0.0937 | 0.4410 | 0.3928 | 0.3379 | −0.4178 | 0.4368 |
| Glucose | −0.4568 | −0.4795 | −o.2779 | −0.3839 | −0.2298 | 0.3864 | −0.0613 |
| Inositol | 0.6037 * | 0.4641 | 0.3635 | 0.6469 ** | 0.8994 ** | −0.7045 ** | 0.4449 |
| Mannitol | −0.5269 * | −0.5735 * | −0.2495 | −0.5232 * | −0.5349 * | 0.7630 ** | 0.6334 * |
| Sucrose | 0.5544 * | 0.6131 * | 0.2536 | 0.528 * | 0.7128 ** | −0.5960 * | 0.1919 |
| Starch | 0.3059 | 0.6559 ** | 0.0383 | 0.1778 | 0.0760 | −0.2074 | −0.0301 |
| Spd | 0.0170 | 0.1233 | 0.0209 | −0.0655 | 0.2079 | −0.3418 | 0.3508 |
| Spm | 0.2843 | 0.1468 | 0.1413 | 0.3141 | −0.0572 | −0.1882 | 0.2791 |
| Ethylene | −0.3180 | −0.4574 | −0.1582 | −0.2197 | 0.1455 | 0.1422 | 0.1709 |

**Table 6.** *Cont.*

| | PDW | RDW | SDW | LDW | LRWC | Leaf Na$^+$ | Leaf K$^+$ |
| | Chl | Chl*a/b* | Car | Proline | Fructose | Glucose | Inositol |
|---|---|---|---|---|---|---|---|
| **Chl** | 1 | | | | | | |
| **Chl*a/b*** | 0.9175 ** | 1 | | | | | |
| **Car** | 0.8799 ** | 0.8425 ** | 1 | | | | |
| **Proline** | 0.3009 | 0.2226 | 0.5777 * | 1 | | | |
| **Fructose** | 0.6258 * | 0.5921 * | 0.6137 * | 0.3294 | 1 | | |
| **Glucose** | 0.0028 | −0.0092 | 0.1048 | 0.0769 | −0.0719 | 1 | |
| **Inositol** | 0.2856 | 0.1636 | −0.0188 | −0.3429 | 0.0517 | −0.2112 | 1 |
| **Mannitol** | −0.2284 | −0.0730 | 0.1936 | 0.3290 | −0.1032 | 0.3068 | −0.5718 * |
| **Sucrose** | 0.2544 | 0.2032 | −0.0606 | −0.2482 | 0.1440 | −0.4065 | 0.7127 ** |
| **Starch** | 0.1918 | 0.1716 | 0.0461 | 0.0171 | 0.0289 | −0.3616 | −0.0280 |
| **Spd** | 0.4439 | 0.1871 | 0.3503 | 0.3610 | 0.4445 | 0.27777 | 0.1192 |
| **Spm** | 0.4941 | 0.6689 ** | 0.5334 * | 0.2186 | 0.4168 | −0.0702 | −0.1733 |
| **Ethylene** | −0.1539 | −0.1539 | −0.2110 | −0.0708 | 0.0352 | 0.3927 | 0.1839 |

| | Leaf K$^+$/Na$^+$ | Root Na$^+$ | Root K$^+$ | Root K$^+$/Na$^+$ | $J_{s\text{-}Na}$ | $J_{s\text{-}K}$ |
|---|---|---|---|---|---|---|
| **Leaf K$^+$/Na$^+$** | 1 | | | | | |
| **Root Na$^+$** | −0.9093 ** | 1 | | | | |
| **Root K$^+$** | 0.8818 ** | −0.8452 ** | 1 | | | |
| **Root K$^+$/Na$^+$** | 0.8818 ** | 0.9092 ** | 0.9412 ** | 1 | | |
| **$J_{s\text{-}Na}$** | −0.8855 ** | −0.8744 ** | −0.8506 ** | −0.8433 ** | 1 | |
| **$J_{s\text{-}K}$** | 0.7972 ** | −0.9117 ** | 0.7995 ** | 0.8296 ** | −0.6905 ** | 1 |
| **Chl** | 0.4103 | −0.4456 | 0.3837 | 0.4542 | −0.6140 * | 0−3224 |
| **Chl*a/b*** | 0.2591 | −0.364 | 0.2492 | 0.2956 | −0.4930 | 0.3098 |
| **Car** | 0.0640 | −0.0521 | 0.0107 | 0.0714 | −0.2929 | −0.0220 |
| **Proline** | −0.1896 | 0.3479 | −0.2804 | −0.3207 | −0.0250 | −0.4719 |
| **Fructose** | 0.2313 | −0.2440 | 0.2476 | 0.1745 | −0.4419 | 0.2214 |
| **Glucose** | −0.1943 | 0.4030 | −0.1204 | −0.10006 | 0.3644 | −0.2708 |
| **Inositol** | 0.6789 ** | −0.7065 ** | 0.8257 ** | 0.8151 ** | −0.6784 ** | 0.6328 * |
| **Mannitol** | −0.8294 ** | 0.8182 ** | −0.832 ** | −0.8107 ** | 0.7788 ** | −0.7032 ** |
| **Sucrose** | 0.5347 * | −0.6429 ** | 0.6046 * | 0.5575 * | −0.5875 * | 0.4866 |
| **Starch** | 0.1360 | −0.2680 | 0.0656 | 0.0606 | −0.2610 | 0.1052 |
| **Spd** | 0.4061 | −0.1614 | 0.3627 | 0.3522 | −0.4692 | −0.0793 |
| **Spm** | −0.1414 | 0.0273 | −0.069 | −0.1130 | −0.2154 | −0.0613 |
| **Ethylene** | 0.0299 | 0.1015 | 0.2341 | 0.1890 | 0.0896 | 0.0033 |

| | Mannitol | Sucrose | Starch | Spd | Spm | Ethylene |
|---|---|---|---|---|---|---|
| **Mannitol** | 1 | | | | | |
| **Sucrose** | −0.4836 | 1 | | | | |
| **Starch** | −0.2035 | 0.5082 | 1 | | | |
| **Spd** | −0.3385 | 0.0933 | 0.1495 | 1 | | |
| **Spm** | 0.0688 | −0.2388 | 0.0366 | 0.0932 | 1 | |
| **Ethylene** | −0.1032 | −0.2325 | −0.5471 * | 0.2777 | −0.3262 | 1 |

## 4. Discussion

Salinity, recognized as one of the most pressing problems affecting the growth and development of plants, can have an important impact on the quantity and quality of yields [41,61,62]. Along with genetic selection, salt tolerance in olive trees can also be improved by using physiological tools [42,63] since numerous species improve both their growth rates and abiotic stress tolerance after the application of PGRs [59,61,63–66]. PGRs have been reported to alleviate the adverse effects of salt stress on physiological and biochemical characteristics of plants and on crop yields [49,67,68]. However, few studies have ever examined the responses of olive plants pretreated with PGRs to salinity.

Our results show that in olive plants of the Picual cultivar not pretreated with PGRs biomass decreased as salinity levels increased, which could be due to lower K$^+$ and greater Na$^+$ concentrations (accumulation of toxic ions) in leaves shown by the high correlations detected between these variables. This observation agrees with the results of other authors who have reported that moderate (100 mM NaCl) and severe (200 mM NaCl) salt stresses are related to a decrease in growth rate in certain olive cultivars [42,69–72]. According to Munns and Tester [73], one of the first responses of plants to salinity is a reduction in leaf growth, which limits the accumulation of toxic ions in leaves and lessens water loss. Salinity at 200 mM NaCl decreased whole plant DW in the Picual plants by 48%. Other authors report whole plant DW reductions of 54% in the cultivar Oueslati [42] and 74% in the cultivar Koronoiki [69,74] state that salinity (200 mM NaCl) severely decreased plant DW in the cultivars Casta Cabra (55.61%), Cornicabra (57.19%), Frantoio (47.06%), Ocal

(35.42) and Picudo (46.08%). Nonetheless, pretreatment with PGRs (GA$_3$, IAA, SA, and Kinetin) helped minimize the adverse effects of salinity on growth. Similar findings were found for other olive cultivars and plant species, in which GA$_3$ application significantly promoted plant growth under salt stress [27,75,76]. According to Yang et al. [77], the effect of plant growth regulators on plants' responses to salinity depends on the species involved. We observed that the application of SA mitigated the inhibition caused by NaCl in the dry weight of olive plants of the cultivar Picual, a similar effect having been found by Methenni et al. [42] for the cultivar Oueslati. However, Aliniaeifard et al. [46] reported for the cultivar Zard that SA did not have any positive effects on growth. Therefore, the effects of SA on the growth of olive plants grown under salt stress is cultivar-dependent. Moreover, it has been shown that exogenous IAA treatments increase plant growth under saline conditions in several commercial crops [78,79], just as we found for the Picual cultivar. Moreover, according to the data obtained in this study, the status of the leaf relative water content (LRWC) in the Picual cultivar was greater than in other Mediterranean cultivars tested for moderate and severe saline stress [71,80].

The olive tree is regarded as a moderately saline-tolerant plant [81], even though the response of plants to saline stress is a cultivar-dependent characteristic [70,75,82]. Mechanisms of salinity tolerance can be classified into three categories: osmotic tolerance, ion exclusion and tissue tolerance [83]. According to Charzoulakis [84], salt tolerance in olive cultivars is associated with the effectiveness of the mechanisms of ion exclusion and the retention of saline ions in roots. In the case of ion exclusion, Na$^+$ is transported to roots, thereby reducing the accumulation of toxic concentrations of Na$^+$ in leaves [83]. Our data indicate that the Picual cultivar not pretreated with PGRs is not able to limit Na$^+$ translocation to actively growing tissues, unlike the other most salinity-tolerant olive cultivars [42,70,75]. However, the Picual cultivar pretreated with IAA and SA were characterized by high accumulations of Na$^+$ in roots and an inhibition of the translocation of this element to leaves. Therefore, of all the pretreatments, those with IAA and SA most favored the ion exclusion mechanism and consequently the salinity tolerance in the Picual cultivar. Similar findings have been reported for the Oueslati cultivar pretreated with SA [42].

Other studies have shown that salinity tolerance is positively associated with a high K$^+$/Na$^+$ ratio [70,80,85]. In fact, plant growth was positively correlated to the K$^+$/Na$^+$ ratio in leaves and roots. It has been shown that under salinity stress, a lower K$^+$/Na$^+$ ratio is provoked by competition between these two ions for binding sites on roots. The entry of the sodium ion into the cytosol through non-selective channels leads to a depolarization of the cell membrane, resulting in K$^+$ leakage and its decrease in cells [19]. Under saline conditions, the K$^+$/Na$^+$ ratio in leaves improved in the Picual cultivar pretreated with IAA and SA. A higher K$^+$/Na$^+$ ratio ensures greater K$^+$ uptake but restricts Na$^+$ ions, which enhances biochemical processes under salinity. Methenni et al. [42] have reported similar results for the cultivar Oueslati pretreated with SA. Additionally, analogous findings have been obtained in other crops after the application of IAA and SA [30,78,86]. Therefore, pretreatment with IAA and SA mitigated salt-induced damage by reducing Na$^+$ accumulation and K$^+$ loss, leading to an improvement in the K$^+$/Na$^+$ ratio in the leaves of the Picual olive cultivar. However, in the pretreatments with GA$_3$ and Kinetin, the improvement in the rate of K$^+$ transport to leaves was not sufficient to compensate for the rate of Na$^+$ transport.

Under saline conditions, the chl($a$+$b$) and carotenoid content of leaves decreased compared to control plants in both non-pretreated and GA$_3$-pretreated plants; on the other hand, in plants pretreated with IAA, SA and Kinetin pigment content was higher than in non-pretreated plants (including under saline conditions). Thus, the carotenoid content was not affected by saline stress. The decrease in photosynthetic pigments provoked by salinity has also been reported for other olive cultivars and tree species [70,82,87]. The reduction in chlorophyll content could be associated with oxidative stress and an increase in reactive oxygen species (ROS) [72]. Similarly, it has been reported that salt stress stimulates the accumulation of ROS, which harms plant tissues as a result of the oxidization of compounds, such as proteins, lipids, carbohydrates, pigments, and nucleic acids [88,89]. Plants can

enhance their tolerance to salt stress by regulating the ROS scavenging [90]. Chl*a*/Chl*b* ratio showed similar values for the IAA, SA and Kinetin pretreatments under both saline and non-saline conditions. The positive effect of IAA and SA pretreatments on the chlorophyll content under salt-stress is consistent with other studies [91–93]. According to Jabeen and Ahmad [94], SA strengthens the antioxidant system and so minimizes the deleterious effects of stress. Bashir et al. [95] suggest that, under stress conditions, carotenoid accumulation in olive trees could be related to metabolic changes occurring during stress adaptation without being directly involved in stress-tolerance mechanisms. In studies of plants under salt stress, Kinetin was able to alleviate oxidative stress by directly or indirectly scavenging more ROS [91,96].

Higher plants can improve their tolerance to salt stress by homeostasis cellular regulation. Organic osmolytes that accumulate in stressful situations include proline, sugars, and sugar alcohols, which are used to regulate osmotic adjustment, maintain turgor, and scavenge ROS [90,97–99], as well as protecting physiological processes against harmful inorganic compounds [100,101]. Proline accumulation is the first response to osmotic stress in plants. This osmolyte has a high hydration capacity, stabilizes subcellular structures, buffers the redox potential, protects the photosynthetic machinery, and, as an antioxidant, acts as a molecular signal that allows for changes in gene expression [95,102–104]. Our data indicate that the Picual cultivar can accumulate proline at 100 mM NaCl, an observation that agrees with previous studies of other olive cultivars and plants [71,81,105–107]. In addition, studies of the salt-tolerant cultivar Chemlali have reported accumulations of proline at 200 mM NaCl [105,107], as occurred in the Picual cultivar pretreated with IAA and SA. Ayaz et al. [70] also detected proline accumulation at 200 mM NaCl in the cultivars Gemlik, Ayvalik and Kilis. Proline accumulation may be linked to protein hydrolysis, a greater expression of enzymes of synthesis, a decrease in degradative enzymes, and less use of this osmolyte due to the use of other osmolytes, such as mannitol and glucose, at higher concentrations [97,108,109].

Glucose, mannitol, fructose, sucrose, starch, and inositol (in decreasing order of abundance) are just five of the sugars present in olive leaves [80]. Mannitol accumulated in the leaves of non-pretreated plants of the cultivar Picual in response to stress at 100 mM NaCl, a finding that confirms previous observations in other olive cultivars [27,71,80,110]. In Picual plants pretreated with PGRs, mannitol accumulated after the addition of both 100 and 200 mM NaCl. Mannitol levels were also higher in plants pretreated with IAA, SA and Kinetin than in non-pretreated plants. These results are confirmed by the correlation obtained between mannitol and $Na^+$ concentrations. It appears that in the Picual cultivar glucose and mannitol synthesis are activated under stress, with mannitol being the most important osmolyte at 100 mM. This osmolyte could help maintain the osmotic balance between cytoplasm and vacuole and contributes to the scavenging of oxygen radicals produced by stress [79]. Moreover, glucose and fructose increased with greater salinity, as has been previously described for other olive cultivars and other plant species subjected to different abiotic stresses [80,111].

Polyamines (PAs) are polycationic compounds with low molecular weights that are present in different cellular compartments. In higher plants, the most common PAs are putrescine (Put), spermidine (Spd) and spermine (Spm) [2–4,112]. These compounds control ion homeostasis, protect photosynthetic pigments, and regulate antioxidant systems and the compounds that stimulate plant abiotic stress tolerance [33,113]. On the other hand, ethylene is a stress hormone and salinity may promote ethylene production and so modulate the activity of the enzymes that regulate the synthesis of this phytohormone [114]. Moreover, Spd and Spm biosynthesis is linked to ethylene production via a common precursor S-adenosylmethionine (SAM) [115,116]. At high NaCl levels (200 mM NaCl), the non-pretreated and IAA-pretreated plants produced more Spd and ethylene than the control plants. In addition, SA-pretreated plants also accumulated Spd at high salt levels, while Kinetin-pretreated plants only accumulated Spm. Therefore, at high NaCl levels (200 mM), Spd would seem to be a good indicator of stress in no-PGR plants and in plants

pretreated with IAA and SA, while Spm could be a good indicator of stress in plants pretreated with Kinetin. Several authors have suggested that Spd levels may be a good indicator of salt tolerance [113,117]. Our results agree with this statement, since in our experiment the plants pretreated with IAA and SA had the highest Spd and Spm levels, respectively, at high salinity, which could be related to their ability to decrease salt-induced damage, prevent $K^+$ loss, reduce $Na^+$ accumulation, and ultimately improve the $K^+/Na^+$ ratio, thereby strengthening the antioxidant systems and protecting the photosynthetic pigments in the Picual cultivar leaves. In other plant species, it has been demonstrated that Spd and Spm can improve growth and reduce salinity-provoked oxidative damage by enhancing antioxidant systems [118,119]. High levels of Spd and Spm perform similar actions in the leaves of the Picual olive cultivar.

## 5. Conclusions

Our results show that the pretreatments of Picual olive cultivar plants with IAA and SA were the most effective of all the studied pretreatments and could significantly improve biomass production under saline conditions. In addition, IAA and SA favored the $Na^+$ exclusion mechanism by reducing $Na^+$ transport from roots to leaves and improved the $K^+/Na^+$ ratio. Soluble sugars increased significantly under salinity, indicating that these two organic osmolytes may play an important role in osmotic adjustment under stress. Specifically, IAA and SA enhanced the accumulation of proline, fructose, and mannitol at 100 and 200 mM NaCl and glucose at 200 mM. The Kinetin pretreatment also favored tissue tolerance through the accumulation of fructose, mannitol (100 and 200 mM), and glucose (200 mM). The IAA and SA pretreatments promoted the accumulation of Spd and Spm, respectively, which could be related to an increase in the antioxidant system, the stabilization of membranes, and greater pigment protection. These results indicate that pretreatments with IAA and SA can alleviate the adverse effects of salt stress in *Olea europaea* Picual cultivar plantlets.

**Author Contributions:** Conceptualization, M.d.P.C. and M.B.; methodology, M.d.P.C., C.A. and M.B.; software, M.M.; validation, M.d.P.C., C.A., M.M. and M.B.; formal analysis, C.A.; investigation, C.A.; resources, M.d.P.C.; data curation, M.d.P.C. and C.A.; writing—original draft preparation, M.d.P.C. and M.B.; writing—review and editing, M.d.P.C. and M.B.; visualization, M.M.; supervision, M.d.P.C.; project administration, M.d.P.C.; funding acquisition, M.d.P.C. All authors have read and agreed to the published version of the manuscript.

**Funding:** This research was supported by the R&D project of Viveros Jarico S.L. funding by the IDEA agency of the Junta de Andalucía, grant number 805 (University of Jaén) Improvement of salinity tolerance in olive by pretreatment with growth regulators.

**Data Availability Statement:** The data presented in this study are available from the corresponding author, María del Pilar Cordovilla, upon request.

**Acknowledgments:** We wish to thank the Central Research Support Services of the University of Jaén for their invaluable help in the growing of plants and quantification of metabolites. We thank Michael Lockwood for his collaboration in the translation of present paper.

**Conflicts of Interest:** The authors declare no conflict of interest.

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
