# Peer review of "Exogenous Application of Indol-3-Acetic Acid and Salicylic Acid Improves Tolerance to Salt Stress in Olive Plantlets (Olea europaea L. Cultivar Picual) in Growth Chamber Environments"

_agronomy, doi:10.3390/agronomy13030647_

Round 1

Reviewer 1 Report

The manuscript deals with an important issue, that of soil salinity, which affects millions of hectares worldwide. The research is sound but there are few points merit attention

1.      Line 30, soil salinity can not be defined only as salt accumulation around plant roots, since saline soils exists also where no plants can be found

2.      L42-44, although the thought is solid, one must have in mind that in order to achieve that, these compounds must have been registered for use in plant species cultivated for food production. To my knowledge, there is no commercially available product registered for use (other than GA3) as foliar application among those you tested here. So this phrase should be re-formed.

3.      L 47 deleted the repeated word this

4.      L53, I believe that the term ‘photosynthetic mechanism’ and not mechanisms is more appropriate

5.      GA3, why not use 3 as a subscript?

6.      L60, improve the sentence by adding K+ concentration

7.      L62, is it ‘photosynthetic assimilation’ or carbon assimilation?

8.      L64, this sentence is not correct. There is much information much more than for the use of SA on the effects of cytokinins and auxins in plants (fruit thinning, adventitious root formation, fruit size stimulation etc). Please re-phrase it, as you probably wanted to emphasize their role in salt tolerance and not a general role.

9.      L69, increases instead of increasing

10.   L71, it is IAA and not AIA

11.   L83, please see remark n. 8

12.   L83-93, English editing is needed (i.e. L86, ‘this fruit tree,…., is widely consumed (not the tree but its products!)

13.   L94, resistance or tolerance?

14.   L94-98, please cite some references for these arguments

15.   L99, replace GAs with GA3

16.   L107, did you use higher night temperature than day temperature? why is that?

17.   L108, the PPFD is quite low compared to natural light This should be mentioned in the title. Furthermore, in the title you claim to have used olive seedlings, but these are not seedlings (according to L104) but plantlets. Please correct and mention in the title that it is ‘a growth chamber experiment’

18.   L114-120 please state what solvent you used for diluting the appropriate mass of each PGR. Furthermore, please state the height of the plants and their age.

19.   L125, not ‘these plants’, but instead the remaining plants

20.   L126-127, what concentration increment of NaCl did you use per day to avoid osmotic shock?

21.   L128, please state that there were six replicates of one plant each

22.   L159-164, why did not you take into account the accumulation of ions in stems too?

23.   L211, what does 1:1 m/v mean in 1.6-diamino-hexane?

24.   L214, use 10 mg ml-1 acetone

25.   L222, can you please state a reference where PAs are measured at 252 x 500 nm? In our laboratory we measure them at λex 365 nm and λem at 475 nm.

26.   L227, for how long?

27.   L239, why did you use an LSD test which is less sensitive than other tests?

28.   L247, Table 1 indicate that root, stem and leaf dry weight as well as LRWC and total plant dry weight were reduced under non saline conditions under the effect of PGRs. Please rephrase and how do you explain that. Furthermore, was the interaction effect significant of the MANOVA analysis, in order to show these differences?

29.   Use dots instead of commas in the numbers in Figures and Tables

30.   The lettering in all Tables should be checked, since there are some mistakes found in Table 2, concerning K concentration.

31.   Be careful with the use of content and concentration. You measured concentration of nutrients and not the content of them.

32.   In many parts of the results there are sentences that fit better to discussion than to results. Please check it out.

33.   Please check your numbers with those reported in the literature and state any observed differences

34.   L511, it is not Picual cultivar(s) (same in other parts of discussion)

35.   L599-601 I believe this is not right, since the nature and mode of action of these compounds determines the need of a plant to synthesize them or not. Please delete it.

36.   In overall, the discussion section is satisfactory, but I urge the authors to compare their numeric results with those of others in order to see possible discrepancies.

37.   The main issue here is that the plant material used is very young plantlets, grown in growth chamber which does not have anything to do with mature productive plants growing outdoors. Although the results are interesting, the title should be more informative and representative. As it is right now is rather misleading. Furthermore, I have not seen anywhere in the results the significance level of the interaction effects presented in tables and graphs. Were all interaction effects significant?

Author Response

Dear reviewer,

Thank you so much for your kind evaluation and positive feedback. We have carefully undertaken all your suggestions/comments and addressed them in the revised manuscript. It is evident that thanks to them the quality of the article will significantly improve. The revised places have been indicated using red color.

Several modifications have been made to the article which we hope you will be pleased with. The presentation of the data has been modified in order to improve it and adapt it to the suggestions made. In the methodology we have completed the description of some methods. In addition, we have completed the conclusions in order to improve it. Moreover, a thorough revision of the English language by a native speaker (Michael Lockwood; mike@walkingcatalonia.net) has been carried out.

In addition, we have made the changes you have suggested and answered the questions raised in the file attached.

Please see the attachmen.

Kind regards,

Reviewer 2 Report

Minor changes are recommended and all of these will be found in detail on the attached manuscript.

Author Response

Dear reviewer, 

Thank you so much for your kind evaluation and positive feedback. We have carefully undertaken all your suggestions/comments and addressed them in the revised manuscript. It is evident that thanks to them the quality of the article will significantly improve. The revised places have been indicated using green color.

We have made the changes you have suggested and answered the questions raised in the file attached in the above box.

Please see the attachmen.

Kind regards,

Reviewer 3 Report

Given below

Exogenous application of indol-3-acetic acid and salicylic acid 2 improves tolerance to salt stress in olive seedlings (Olea euro-3 paea L. cultivar Picual)

·       Line 3-35. “Currently, worldwide salinized land occupies a total of 932.2 Mha on all continents and in Europe is mainly concentrated in the Mediterranean area”.

No need to write “on all continents” when “worldwide” is already mentioned.

The 2nd part of the sentence should be restructured as it describes the part of Europe where saline land is concentrated. Please check the sentence for a better understanding.

·       Line 39-44.  Gene-editing technologies, genome database information, and transgenesis [8–10], in addition to seed priming and foliar spraying (techniques that enhance germination and growth development by activating various physiological and biochemical processes) with compounds such as plant growth regulators (PGRs) [11–15], could all improve abiotic stress tolerance and ensure sustainable, highly nutritional food derived from herbaceous and perennial crops.

Do you mean these “Gene-editing technologies, genome database information, and transgenesis” are the only available technologies for stress tolerance, or are you mentioning these as the most recent technologies? Please check.

·       Line 48 “this PGR mitigates.” Please check and delete “this.”

·       Line 51-52. Increases antioxidant activity and produces greater accumulations of osmolytes…..

Please check the sentence. Wouldn’t it be better as it “increases antioxidant activity and accumulations of osmolytes…”

·       Line 69- “as well as increasing the number of nodules for better nitrogen fixation”

“..Increases the number of……” Please check.

·       Line 71. “…AIA and KIN have been”

AIA? Please check. Do you mean IAA?

·       Line 91. The use of ecologically and environmentally friendly strategies to achieve more.”

Ecological and environmentally friendly strategies? Please check.

·       Line 94. Resistance to salinity can be achieved by stimulating (i) water content for better growth,

Water content for better growth? Please specify. It improves the available water content to plant roots? Or increases the water content in the plant.? It is unclear here.

·       Line 106.  under the following conditions:

Following? Please check.

·       Line 104. Mist-rooted Picual olive cuttings

A number of cuttings should be mentioned here.

·       Line 141. dry weight of the whole plants and organs,

organs? Please specify.

·       Line 148 water and dried with filter paper

Dried between the layers of filter paper? Please specify.

·       Part of the plant material (roots, stems, and leaves) from both control and PGR-treated plants (with salt and without salt) was dried at 70°C for 72 h in a forced-air oven, and the dry weight (DW) of the different organs was determined.

Parts of the plant material?

Please use the same term throughout the document, either plant parts or organs.

Use “were” in place of was dried at…..

·       Line 170 the supernatant being used for the quantification of the proline

Being used, or was it used? Please check.

·       Line 245….Data for roots, stem, and leaf dry weight, as well as for whole plant dry weight, regarding the response to salinity in (i) olive seedlings pretreated with PGRs and (ii) not pretreated (no-PGR) are given in Table 1 and Figure 2.

There is no need for this sentence. Provide the results and then mention the specific figure or table to look for. Please check the results.

Author Response

Dear reviewer, 

Thank you so much for your kind evaluation and positive feedback. We have carefully undertaken all your suggestions/comments and addressed them in the revised manuscript. It is evident that thanks to them the quality of the article will significantly improve. The revised places have been indicated using blue color.

Several modifications have been made to the article which we hope you will be pleased with. The presentation of the data has been modified in order to improve it and adapt it to the suggestions made. In the methodology we have completed the description of some methods. In addition, we have completed the conclusions in order to improve it. Moreover, a thorough revision of the English language by a native speaker (Michael Lockwood; mike@walkingcatalonia.net) has been carried out.

Best regards,

Reviewer 4 Report

Salinity is one of the key abiotic stresses, which severely affects crop productivity in arid and semiarid environments. Besides genetic selection, physiological tool, such as application of PGRs, can also improve the abiotic stress tolerance of crops. In this work,

Five PGRs, gibberellic acid (GA3), indole-3-acetic acid (IAA), salicylic acid (SA) and Kinetin (KIN) were applied on olive plant, Picual, to assess their ability to alleviate the salinity stress. The results showed that pretreatments with IAA and SA might be a highly effective way of increasing salt tolerance in olive seedlings.

The manuscript is well structured, organized and written. I think it is suitable for Agronomy.

There are a few minor errors.

P71, AIA should be IAA.

P107, 20–25°C day-night temperature may be a little low, pls check it.

P568, is a punctuation missing between sucrose starch ?

Author Response

Dear reviewer, 

Thank you so much for your kind evaluation and positive feedback. We have carefully undertaken all your suggestions/comments and addressed them in the revised manuscript. Moreover, a thorough revision of the English language by a native speaker (Michael Lockwood; mike@walkingcatalonia.net) has been carried out.

Best regards,

Round 2

Reviewer 1 Report

The manuscript has been revised and reached a good level.

My one remark is to add how the tissue was dried and how the chlorophyll was calculated per dry weight basis, since fresh material was used for the extraction (the same stands for all measurements where fresh weight was used and the results are expressed per dry weight)

Author Response

Dear Reviewer,
thank you for your interest and recommendations.
Regarding your comment, the methodology used for the determination of the different parameters was chosen according to the latest published articles and considering that they offer reliable data. 
In relation to the expression of the data, the authors who work on salinity present them frequently in dry weight, as this way the effect of saline stress and the differences between treatments are better detected.
For the transformation of fresh weight into dry weight, for each pre-treatment with growth regulators and NaCl level, 3 tissue samples were used; first the fresh weight was determined and after drying the dry weight was determined. For this, the same amount of fresh plant material was used as for the determination of the different parameters. This allowed the conversion from fresh weight to dry weight. The plant material was dried at 70°C for 72 h in a forced-air oven and the dry weight (DW).
Other authors who use the same procedure to make the parameter determination in fresh weight and then express it in dry weight are:

-Roshdy, E.A.; Alebidi, A.; Almutairi, K.; Al-Obeed, R.; Elsabagh, A. The effect os salicylic acid on the performances of salt stressed satrawberry plants, enzymes activity, and salt tolerance index. Agronomy 2021, 11, 775. https://doig.org/103390/agronomy11040775.

-Shin, Y.K.; Bhandari, S.R.; Jo, J.S; Song, J.W.; Cho, M.C.; Yand, E.Y., Lee, J.G. Response to salt stress in lettuce: Changes in chlorophyll fluorescence parameters, phytochemical contents, and antioxidant activities. Agronomy, 2020, 10, 1627. https://doig.org/103390/agronomy10111627.

-Taha, R.S.; Seleiman, M.F., Alotaibi, M.; Alhammad, B.A.; Rady, M.M.; Mahdi, A.H.A. Exogenous potassium treatments elevate salt tolerance and performances of Glycine max L. by boosting antioxidant defense system under actual saline field conditions. Agronomy 2020, 10, 1741. https://doig.org/103390/agronomy10111741.

-Al-Sahamsi, N.; Hussain, M.I.; El-Keblawy A. Physiological responses of the xerohalophyte Suaeda vermiculata to salinity in its hyper-arid environment. Flora 2020, 273, 151705. https://doig.org/10.1016/j.flora.2020.151705.

-Methenni, K.; Ben Abdallah. M.; Nouairi, I.; Smaoui. A.; Ben Ammar, W.; Zarrouk, M.; Ben Youssef, N. Salicylic acid and calcium pretreatments alleviate the toxic effect of salinity in the Oueslati olive variety. Sci. Hort. 2018, 233, 349-358. https://doig.org/10.1016/j.scienta.2018.01.060.

-Aparicio, C.; Urrestarazu, M.; Cordovilla, M.P. Comparative physiological analysis of salinity effects in six olive genotypes. HortScience 2014, 49, 901-904. https://www.academia.edu/26044229/Comparative_Physiological_Analysis_of_Salinity_Effects_in_Six_Olive_Genotypes.

The article has again been revised by the native English translator (Michael Lockwood; mike@walkingcatalonia.net).

King regards,